# Research on desiliconization of brown corundum fly dust and bauxite based on roasting-alkali leaching method

**Nian Liu[1,2], Chaoyi Chen[1,2]\*, Junqi Li[1,2], Dong Liang[1,2]**

**1** School of Materials and Metallurgy, Guizhou University, Guiyang China, **2** Guizhou Key Laboratory of Metallurgical Engineering and Process Energy Conservation, Guiyang, China

\* czchen@gzu.edu.cn

## Abstract

The raw material for smelting brown corundum is high-quality bauxite. As the quality of bauxite decreases, the main impurity, silicon content, increases, which affects the product quality and smelting energy consumption. Additionally, the smelting process produces a significant amount of brown corundum fly dust (BCFD) with a low utilization rate, resulting in resource wastage. In order to utilize low-quality bauxite and BCFD, this article proposes the method of "roasting-alkali leaching." Through mixed desilication, the impact of alkali leaching factors on desilication is investigated, analyzed the desiliconization mechanism of BCFD, and established the kinetics of the desilication reaction. Results indicated that the optimal conditions for mixed desilication are BCFD/bauxite mass ratio of 1:6, desilication temperature of 95°C, desilication time of 30 min, alkali concentration of 110 g/L, and liquid-solid ratio of 10:1, achieving a desilication rate of 60.90%. The Al-Si ratio (A/S) of the concentrate increases from 5.33 to 11.72, meeting the requirements for brown corundum smelting raw materials. The desilication reaction follows a solid shrinkage core model, with a kinetic equation of $1 - 2/3\alpha - (1-\alpha)^{2/3} = 15.50 \exp[-29299/(RT)] \cdot t$, and an apparent activation energy of 29.30 kJ/mol. The synergistic mechanism involves fine particles of the BCFD adsorbing on the surface after mixing with the bauxite, increasing the mineral surface area and the activity of silicon, thereby accelerating the reaction rate.

## 1. Introduction

Bauxite, as the main raw material for the production of brown corundum, is also used in the production of alumina, refractory materials, abrasive materials, chemical products, and high-alumina cement [1,2]. Bauxite reserves in China are abundant, but most are dominated by monohydrate bauxite, characterized by high alumina [3], high silica [4] and low iron [5]. Bauxite is mainly used for alumina production, which has had a significant impact on the brown corundum industry that primarily relies on high-grade bauxite as its main material [6]. With industrial development, the grade of bauxite is decreasing, and the content of various impurities is increasing, especially $SiO_2$. Therefore, pretreatment of raw materials to reduce the main impurity silicon is currently a research focus.

**Data availability statement:** The underlying data is available in the Open Science Framework (OSF) repository at (https://osf.io/tfdr5/). DOI: 10.17605/OSF.IO/TFDR5.

**Funding:** This work was financially supported by the National Natural Science Foundation of China (52164017,U23A20610,52074096,52274260).

**Competing interests:** The authors have declared that no competing interests exist.

Brown corundum, with its high hardness and toughness, is used in the fields of abrasives and refractory materials [7], and is an indispensable material in industry and production [8], However, a large amount of BCFD is generated during the smelting process, whose main components are $SiO_2$, $Al_2O_3$, and $K_2O$, and it also contains small amounts of rare elements Ga and Rb [9,10]. The discharge and accumulation of a large amount of untreated dust pose a serious threat to the ecological environment. Therefore, the rational use of BCFD can obtain valuable resources and protect the environment.

Effective results have been achieved in the desilication of low-grade bauxite, with the main methods including biological desilication [11], flotation desilication [12] and roasting-alkali leaching desilication [13].Teng [14] used biological methods to treat magnesite, The experimental conditions were a bacteria concentration of 10 g/L, a leaching temperature of 30°C, and a leaching time of 3d. and silicate bacteria reduced the $SiO_2$ content in the mineral from 4.61% to 2.56%,Wang [15] cultivated two efficient strains to remove silicon from high-silica bauxite, The experimental conditions were two concentrations of 9.1 and 9.64, leaching time and temperature were 30°C and 7 d respectively, and the Al-Si ratio of the desilicated concentrate reached above 9, meeting the requirements for alumina production in the Bayer process. Biological desilication uses microorganisms to decompose and dissolve silicon in minerals, but the cycle is long. Zhou [16] used reverse flotation on bauxite, when the concentration of the agent is 0.4 mmol/L and pH = 5, reducing the $SiO_2$ content in the desilicated concentrate from 14.15% to 10.35%. Flotation desilication can improve the Al-Si ratio, but the addition of organic matter affects the leaching of alumina. Xu [17] added red mud to the aluminate solution for desilication, When the alkali concentration was 110 g/L, the temperature was 95°C, and the reaction time was 2h, achieving a desilication rate of 89% and increasing the Al-Si ratio to 7.5. Li [18] used the roasting-alkali leaching process on high-silica bauxite, with an optimal roasting temperature and time of 1050°C and 30 min, desilication temperature and time of 95°C and 30 min, with the silica leaching rate of 88% for the desilicated minerals. Roasting-alkali leaching desilication is the most widely used method at present, but the composition and mineral phase structure are complex, and how to improve efficiency is a research focus.

The treatment methods for BCFD mainly include acid and alkali methods. Wen [19] treated BCFD with a mixture of sulfuric acid and hydrofluoric acid, and when their concentrations were 1.5 and 6.4 mol/L, respectively, the gallium leaching rate was 91%. Ding [20] treated the BCFD with ultrasound, and when the sulfuric acid concentration was 25 wt% and the liquid-solid ratio was 1.2, the temperature and time were 60°C and 15min, the gallium leaching rate was 82.56%. Xiong [21] proposed a pollution-free process route for recovering brown corundum dust, under the experimental conditions of a liquid-solid ratio of 20:1, a desilication temperature and time of 60°C and 15 min, with the resulting product having an Al-Si ratio of 4.95 and a $SiO_2$ content of 57.57%. Both acid and alkali methods face challenges; the acid method has a low extraction rate and severe corrosion, while the alkali method can recover alumina-silicate minerals but still requires further research.

To solve the problems of low Al-Si ratio in bauxite and the difficulty in treating BCFD, this paper proposes a synergistic desilication route of BCFD and bauxite roasting-alkali leaching. It explores the synergistic desilication mechanism of the BCFD, aiming to improve the grade of bauxite and solve the utilization rate of BCFD. The synergistically desilicated ore can be used as a raw material for brown corundum smelting, expanding the range of raw material choices.

## 2. Materials and methods

### 2.1 Materials

The bauxite and brown corundum fly dust (BCFD) used in the experiment were sourced from a Guizhou abrasive company. Their chemical compositions and phases are as shown in

Table 1 and Fig 1, respectively. The main phases of the bauxite are diaspore, quartz, anatase and kaolinite; the primary phases of the BCFD are alumina, potassium sulfate, and silicon dioxide, Notably, the silica is amorphous, exhibiting alkali solubility. The BCFD contains 20.23% alumina, which has value for recovery and reuse. The alkali solution was prepared using analytical grade reagents from Aladdin Reagent Company.

**Table 1. Main components of bauxite and BCFD (wt.%).**

|  | $Al_2O_3$ | $SiO_2$ | $Fe_2O_3$ | $TiO_2$ | CaO | MgO | S | $K_2O$ | A/S |
|---|---|---|---|---|---|---|---|---|---|
| Bauxite | 74.39 | 8.70 | 2.02 | 4.12 | 0.23 | 0.21 | 1.01 | 0.27 | 8.55 |
| BCFD | 20.23 | 36.38 | 4.01 | 1.07 | 0.54 | 1.32 | 3.39 | 17.90 | 0.56 |

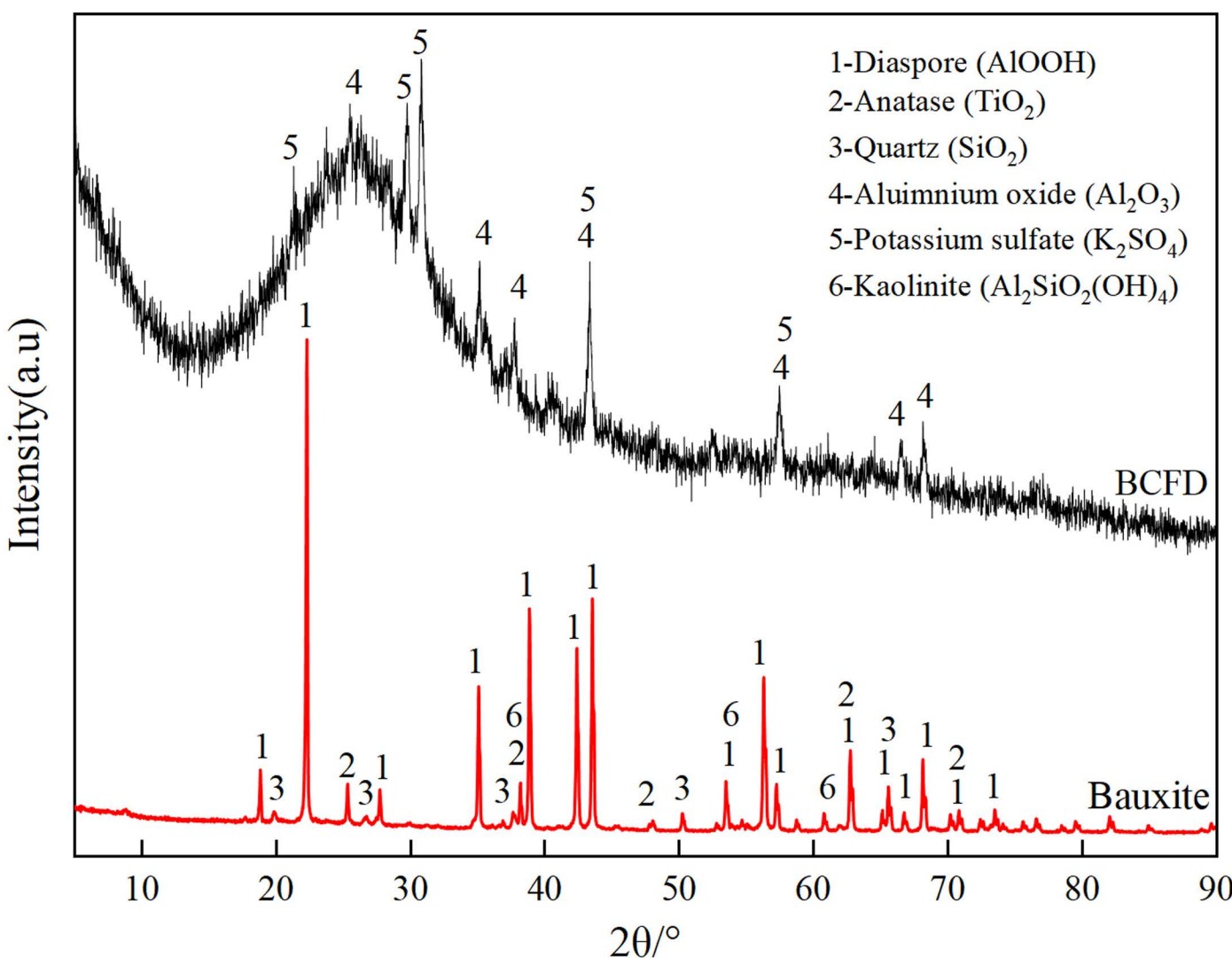

**Fig 1. Bauxite and brown corundum fly dust XRD pattern.**

## 2.2 Experimental methodology

The bauxite was roasted in a muffle furnace at 1050°C for 30 min. After cooling to room temperature, it was subjected to alkali leaching with different ratios of BCFD. The leaching conditions were as follows: the alkali solution had a $Na_2O$ concentration of 110 g/L, a liquid-to-solid ratio of 10, leaching temperature of 95°C, and leaching time of 30 min. After alkali leaching, vacuum filtration was performed using a vacuum filtration apparatus. The filtered samples were washed with hot water three times or more, dried at 80°C for 12 hours, and subsequently subjected to grinding and analysis. The process flow described above in Fig 2.

## 2.3 Analysis methodology

The mineral composition of the samples was determined using an X-ray diffractometer (XRD, SmartLab SE) with CuKα1 radiation. The main components of the samples were tested by chemical analysis, and trace elements in solid samples were measured using an X-ray fluorescence spectrometer (XRF, XRF-1800). The microstructure of the roasted products and the desilicated concentrate was characterized using a scanning electron microscope (SEM, Sigma 300), and the elemental distribution of the roasted products was analyzed using an energy dispersive spectrometer (EDS, X-Max 50). Additionally, the desilication rate, alkali consumption, and aluminum loss of the BCFD mixed with bauxite for desilication were calculated according to formulas (1), (2), and (3).

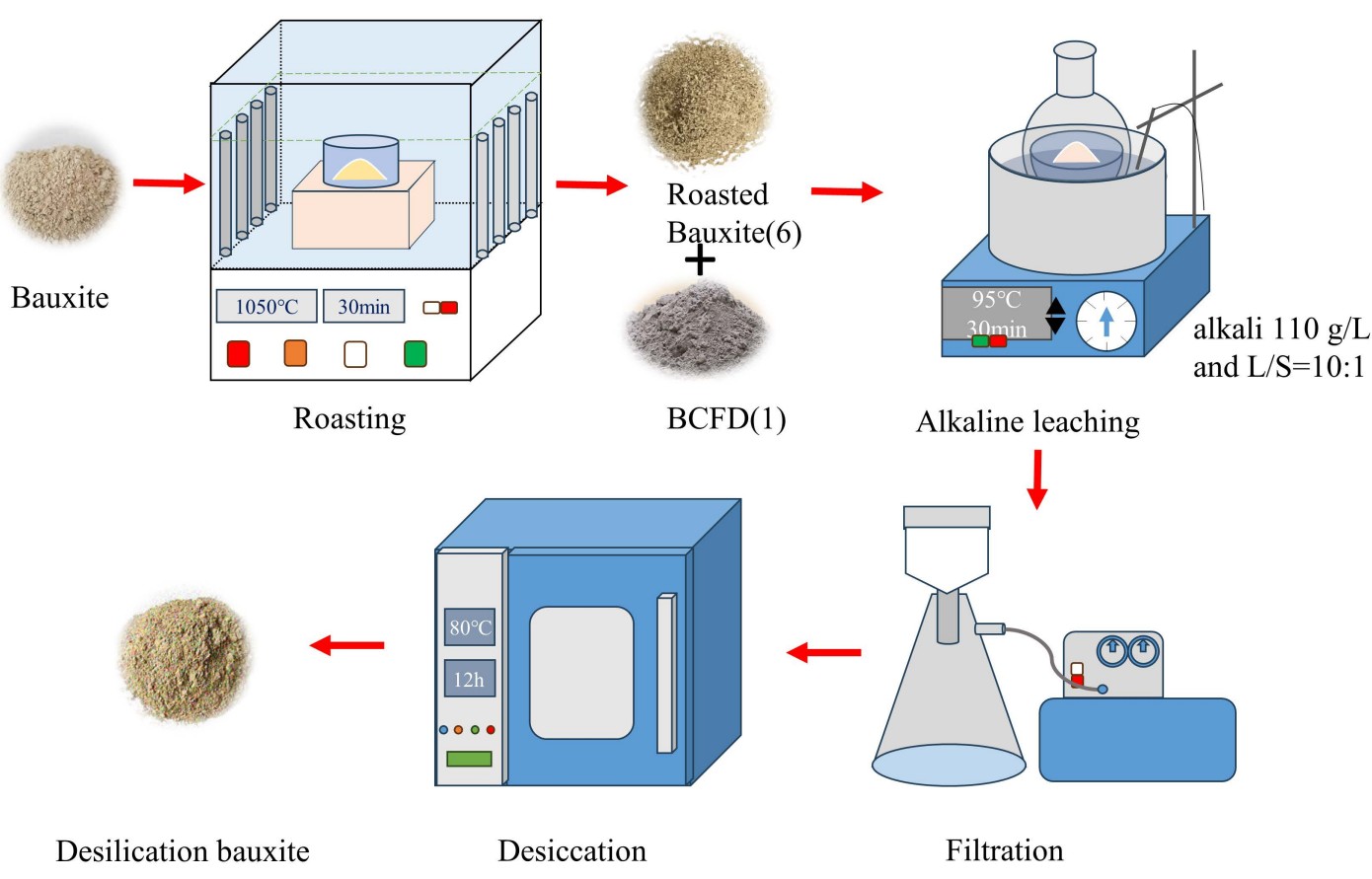

**Fig 2. Flowsheet of the experimental process.**

$$\eta_1 = \frac{MF - M_1 F_1}{MF} \tag{1}$$

where $\eta_1$ is the desilication rate and M is the mixed ore content; F is the mixed ore $SiO_2$ content, $M_1$ is the mixed ore content after desilication; and $F_1$ is the mixed ore $SiO_2$ content after desilication.

$$\alpha = \frac{SV - S_1 V_1}{SV} \tag{2}$$

where $\alpha$ is the alkali consumption, S is the initial alkali content; V is the initial alkali volume, S1 is the post desilication alkali content; and $V_1$ is the post desilication alkali volume.

$$\theta = \frac{mA - m_1 A_1}{mA} \tag{3}$$

where $\theta$ is the aluminium loss rate and m is the mixed ore content; A is the $Al_2O_3$ content of the mixed ore, $m_1$ is the content of the mixed ore after desilication; $A_1$ is the $Al_2O_3$ content of the mixed ore after desilication.

## 3. Results and discussion

### 3.1 Desilication process

**3.1.1 The impact of raw material ratio.** The impact of raw material ratio on the desilication of mixed ore is shown in Fig 3. From Fig 3(a), it can be seen that as the proportion of bauxite roasted ore increases, the A/S of the mixed ore increases, the alumina content increases, and the silicon dioxide content decreases. From Fig 3(b), the desilication effect of the dust from individual raw materials and the bauxite calcination ore is relatively poor, with desilication rates of 42.79% and 39.81%, and A/S ratios of 1.07 and 10.79, respectively. When the proportion of bauxite roasted ore increases, both the desilication rate and A/S increase, reaching a maximum at a raw material ratio of 1:6, with a desilication rate and A/S of 60.90% and 11.72, respectively. As the ratio continues to increase, both the desilication rate and A/S decrease. Therefore, it indicates that adding an appropriate amount of BCFD can improve the desilication effect [22].

Fig 3(c), it can be seen that the alkali consumption for desilication of bauxite calcination ore alone is relatively high. As the proportion of bauxite increases, both the aluminum loss and alkali consumption in the filtrate also increase. Fig 3(d), it is clear that the diffraction peak intensity of the desilicated concentrate from BCFD is lower, indicating poor crystallinity and a lower desilication rate. The desilicated concentrate from bauxite shows diffraction peaks of the quartz phase, leading to a lower content of desilicated silicon dioxide. As the raw material ratio increases, the diffraction peak intensity of corundum increases. Based on the relationship between desilication rate, alkali consumption, and aluminum loss for different ratios, the optimal raw material ratio is determined to be 1:6.

**3.1.2 Effect of alkali concentration and liquid-solid ratio.** The effects of different alkali concentrations and liquid-solid ratios on the desilicated concentrate are shown in Fig 4. Form Fig 4(a), with the increase in alkali concentration, the A/S and desilication rate of the desilicated concentrate gradually increase. At an alkali concentration of 80 g/L, the A/S of the desilicated concentrate increases from 5.33 to 8.38, and the desilication rate is 45.17%. When the alkali concentration is 110 g/L, the A/S of the desilicated concentrate is 12.10, and the desilication rate reaches 60.90%. Continuing to increase the alkali concentration results in a decrease in both the desilication rate and A/S. This is because increasing the

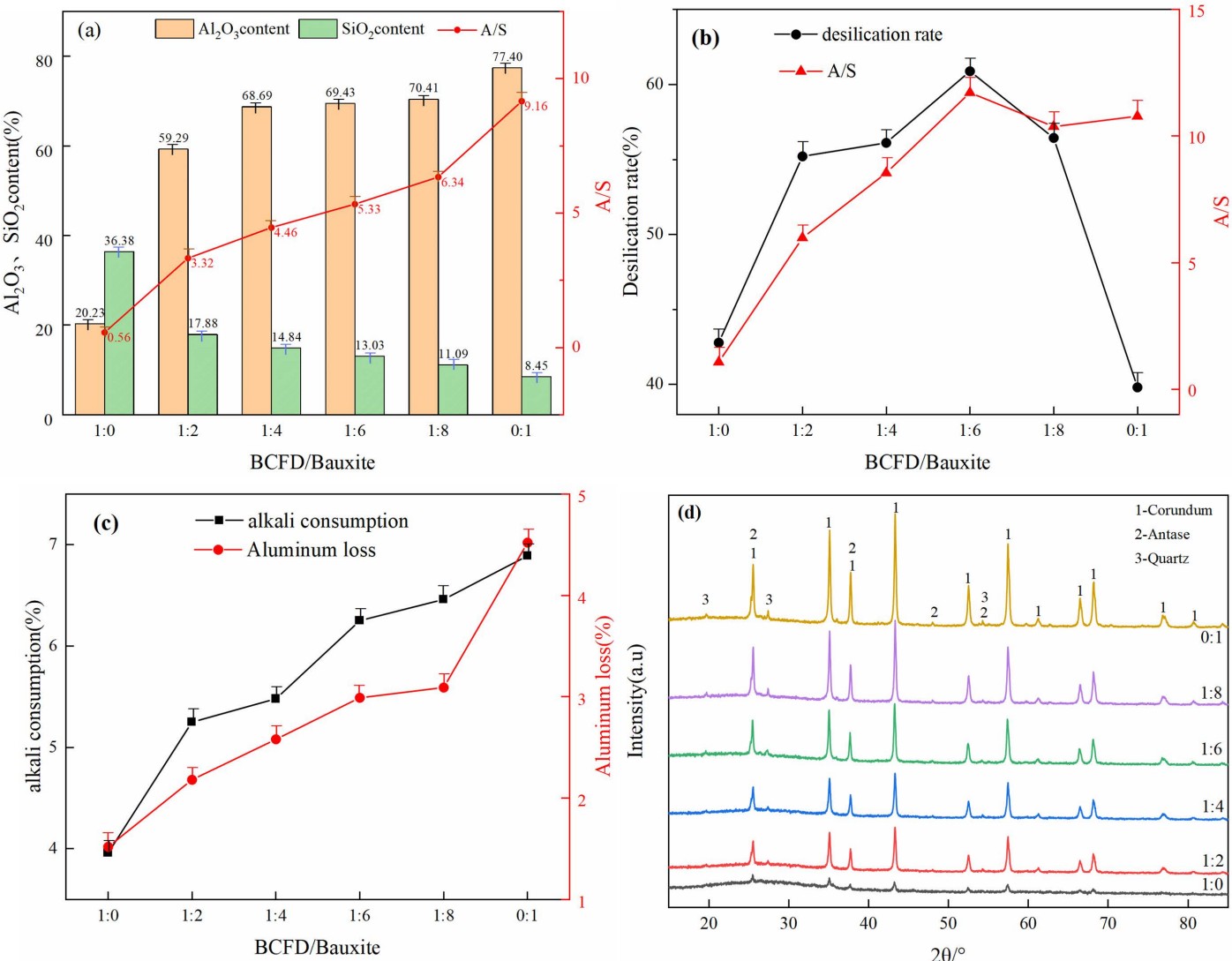

**Fig 3. Effect of raw material ratios on desilication of mixed ores** (a) A/S of raw material ratios; (b) Desilication rate and A/S; (c) Alkali consumption and aluminum loss (d) XRD of desilicited in different ratios (desilication conditions: 95°C, 30 min, 110 g/L, 10:1).

alkali concentration increases the activity of OH⁻ in the system, which is beneficial for the dissolution of amorphous $SiO_2$, forming $NaSiO_3$ [2,23]. The A/S of the desilicated concentrate has reached the requirements for industrial production of brown corundum. According to Fig.4(b), An excessively high alkali concentration can easily cause aluminum loss, resulting in a lower A/S ratio. This indicates that although the increase in alkali concentration increases the silicon content entering the solution, it also tends to increase the rate of aluminum loss. Therefore, it is comprehensively judged that an alkali concentration of 110 g/L is a better desilication alkali concentration.

As shown in Fig 4(c) and (d), increasing the liquid-to-solid ratio initially enhances the desilication rate and A/S ratio of the desilicated concentrate, which then stabilizes, while alkali consumption and aluminum loss gradually increase. At a ratio of 6:1, the desilication rate and A/S ratio of the desilicated concentrate are 52.61% and 9.12 respectively, with alkali

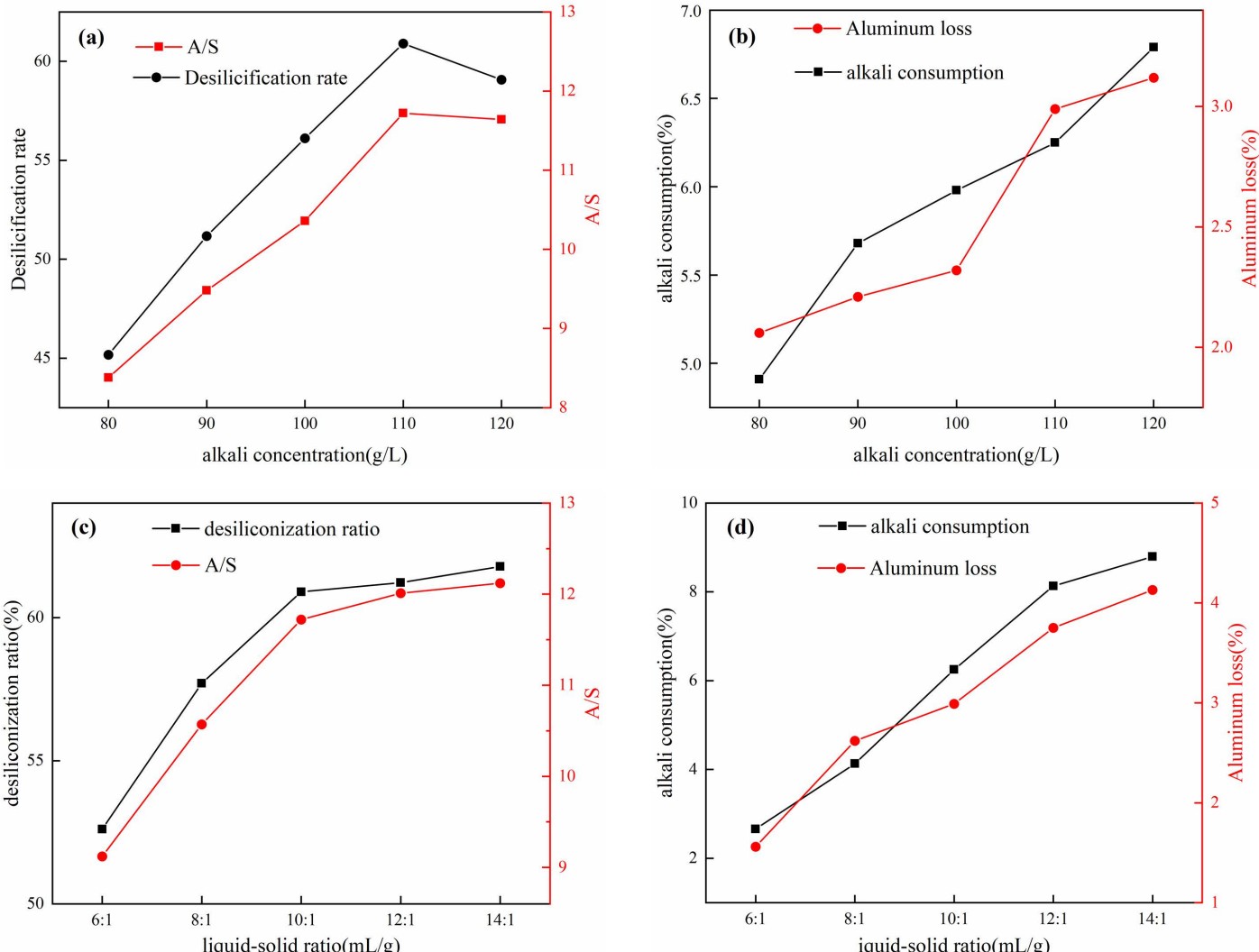

**Fig 4. Effect of alkali concentration on mixed desilication** (a) desilication rate and A/S(b) alkali consumption and aluminum loss; Effect of liquid-solid ratio on mixed desilication (c) desilication rate and A/S(d) alkali consumption and aluminum loss.

consumption and aluminum loss at 2.66% and 1.56%. With a ratio of 10:1, the desilication rate and A/S ratio reach 60.90% and 11.72, while alkali consumption and aluminum loss increase to 6.25% and 2.99% respectively. At this point, further increasing the liquid-to-solid ratio reduces the rate of increase in desilication rate and A/S, but alkali consumption and aluminum loss continue to rise. A higher liquid-to-solid ratio reduces the solute mass, decreases viscosity in the alkali reaction system, accelerates reaction rates, and promotes the dissolution of active silica [24]. Therefore, a ratio of 10:1 is chosen as the optimal liquid-to-solid ratio.

**3.1.3 Effect of desilication temperature and time.** The influence of different desilication temperatures and times on the desilicated concentrate is shown in Fig 5. According to Fig 5(a), (b), with the increase in desilication temperature, the desilication rate and A/S ratio of the mixed desilicated concentrate initially increase and then decrease, while alkali consumption and aluminum loss both increase. At a temperature of 95°C, the desilication rate and A/S ratio of the desilicated concentrate are 60.90% and 11.72 respectively, with

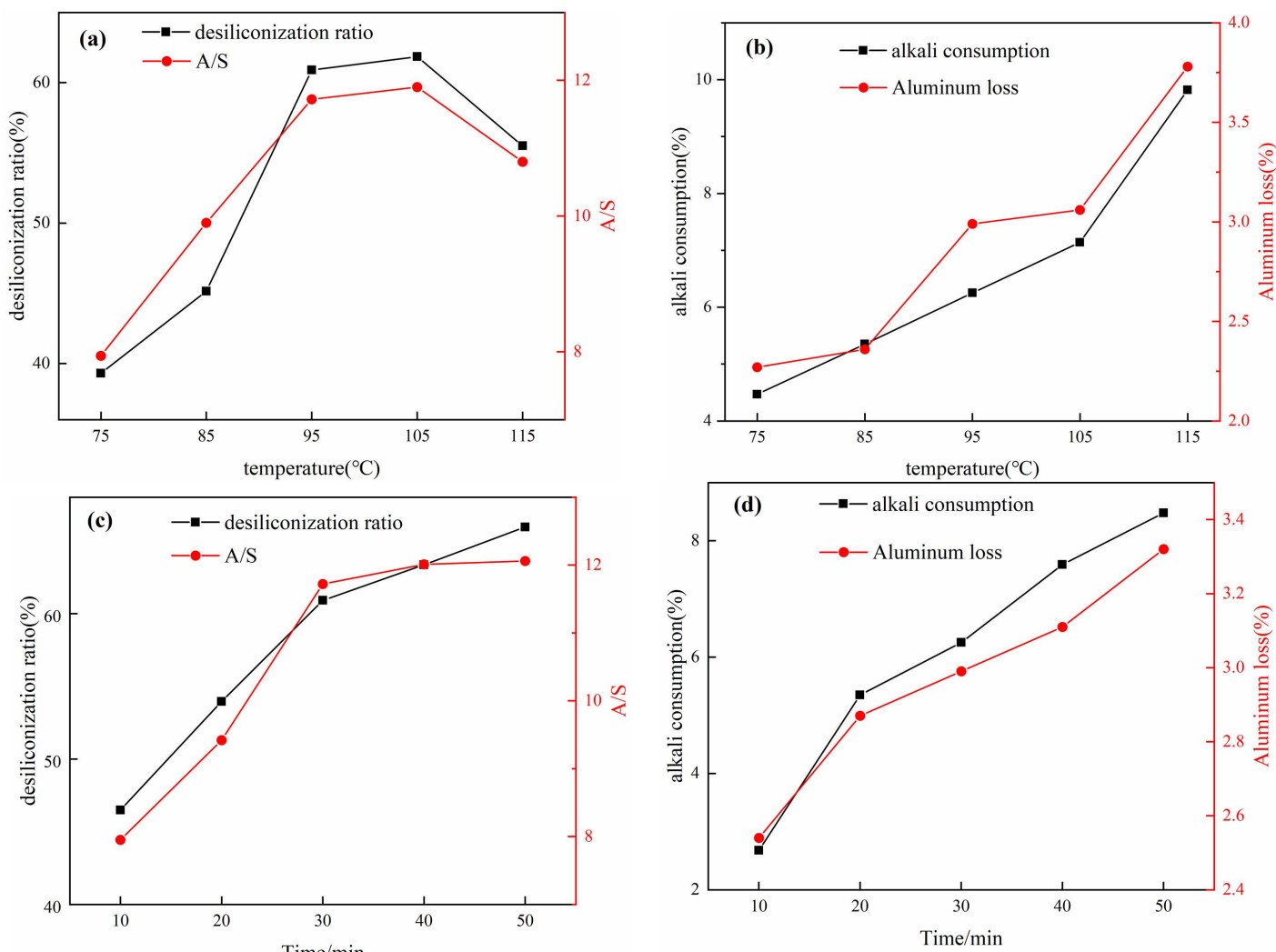

**Fig 5. Effect of desilication temperature on hybrid desilication** (a) desilication rate and A/S(b) alkali consumption and aluminum loss; effect of desilication time on hybrid desilication (c) desilication rate and A/S (d) alkali consumption and aluminum loss.

alkali consumption and aluminum loss at 6.25% and 2.99%. When the temperature is raised to 115°C, the desilication rate and A/S ratio decrease to 55.50% and 10.80, while alkali consumption and aluminum loss increase to 9.82% and 3.78%. Increasing the temperature reduces the viscosity of the system in the alkali solution, enhances the reaction rate, accelerates the dissolution of active silica, speeds up the precipitation of sodium aluminum-silicate, and improves the desilication rate. As the temperature rises, the alkali solution reaches its boiling point, leading to evaporation and a reduction in the liquid-to-solid ratio, increased system viscosity, which is unfavorable for the dissolution of silica, and an increase in the solubility of alumina [25,26]. Then was selected as the optimum desilication temperature. Therefore, 95°C is chosen as the optimal desilication temperature.

Fig 5(c), (d) with increasing desilication time, the desilication rate and A/S ratio of the desilicated concentrate initially increase and then stabilize. At 30 minutes, the desilication rate and A/S ratio are 60.90% and 11.72 respectively, with alkali consumption and aluminum

loss at 6.25% and 2.99%. After 30 minutes, the desilication rate and A/S ratio of desilicated concentrate slowly increase and gradually stabilize. To control the leaching of silica dioxide, reduce alkali consumption, and minimize economic losses, 30 minutes is chosen as the optimal desilication time.

In summary, the better factors for BCFD/bauxite synergistic desilication were a raw material ratio of 1:6, desilication temperature and time of 95°C and 30 min, alkali concentration and liquid-solid ratio of 110 g/L and 10:1, and desilication rate and A/S of 60.90% and 11.72 under this condition. The desiliconization rate of bauxite after roasting is 39.12%, and the desiliconization effect of bauxite after roasting is mixed with BCFD to desilication, and the desilication rate is 60.90%.

## 3.2 Process discussion

**3.2.1 Morphological analysis.** Physical phase analyses of before and after roasting of bauxite ore, before and after desilication of BCFD, and before and after desilication of a mixture of BCFD/bauxite ore, were carried out to investigate the migration of elements. The result is shown in Fig 6.

As shown in Fig 6(a), after roasting of bauxite, diaspore is transformed into transition state alumina, and kaolinite decomposes into reactive silica and spinel phases, causing the mineral to crack and become loose, which promotes the combination of alkali solution and mineral, accelerates the reaction rate, and increases the desilication rate [27]. Fig 6(b) shows the desilicated concentrate of bauxite, where active silica is removed from the mineral, reducing the particles combined with aluminum and silicon, and making the ore loose. Fig 6(c) mainly consists of fine round particles and a small amount of blocky ones, where the round particle spheres are mainly quartz existing in the form of silica, and the blocky distribution is mainly potassium sulfate, Fig 6(d) the desilicated of BCFD, where the silicon content is reduced, and there is still some residual non-reactive silica that did not participate in the reaction, existing in the form of quartz, which is difficult to leach out under normal pressure alkali leaching. Fig 6(e) the BCFD/bauxite ore, where the BCFD particle spheres are concentrated around the bauxite, with a few small spheres integrated into the blocky bauxite particles. This increases the active silica content on the bauxite surface, which is beneficial for subsequent alkali leaching desilication [28]. Fig 6(f) the BCFD/bauxite desilicated ore, after desilication, the large particles combined with aluminum and silicon are reduced, holes and cracks appear on the particle surface, the ore structure becomes loose, the silicon content is lower, and a small part of titanium exists in the form of titanium dioxide.

**3.2.2 Desilicition characterization.** Table 2 shows the chemical composition of the BCFD/bauxite and desilicated concentrate, indicating that the silicon dioxide content decreased from 13.03% to 6.43%, the alumina content increased to 75.35%, and the A/S ratio improved from 5.33 to 11.72, meeting the requirements for brown corundum smelting raw materials.

Fig. 7 shows the XRD of the desilicated concentrate, with the main phases being corundum, quartz, and anatase. Compared to the raw ore, the silicon-containing diffraction peaks are reduced, with residual non-active silica quartz phases that cannot be eliminated during the desilication process, as well as anatase that participated in the reaction [29].

In order to explore the synergistic effect of BCFD, the pore size distribution test was carried out on the mixed ore before and after desillication, and the changes before and after desiliconization were analyzed. This is shown in Fig 8. The results show that after the addition of BCFD for alkali leaching and desilication, the pore size increases, and the surface area increases, and the BCFD particles are smaller, which is enriched on the bauxite particles, and the interaction occurs, which increases the desiliconization rate.

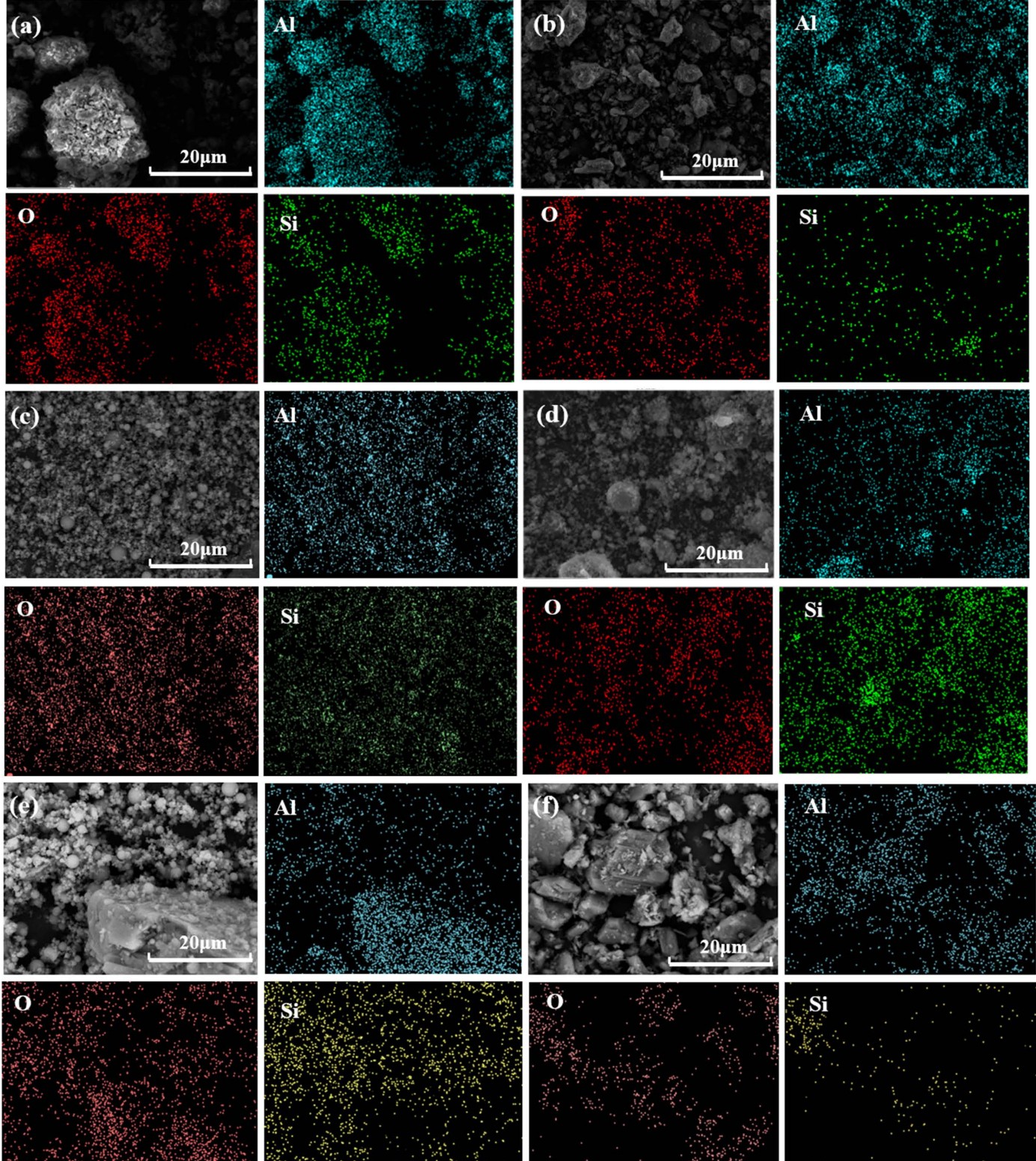

**Fig 6. Morphology for different conditions.** (a) bauxite roasted ore, (b) bauxite desilication ore, (c) BCFD raw ore, (d) BCFD desilication ore, (e) BCFD/ bauxite ore, (f) BCFD/ bauxite desilication ore.

**3.2.3 Desilication dynamics.** Roasting can convert the kaolinite in bauxite into active silica, which reacts with the alkali solution and removes the silica from the mineral. The mechanism of alkali solution desilication is actually a liquid-solid reaction between the mineral and the alkali solution. The leaching process conforms to the shrinking model, mainly

**Table 2. Main components of BCFD/bauxite and desiliconised concentrates (wt.%).**

|  | $Al_2O_3$ | $SiO_2$ | $Fe_2O_3$ | $TiO_2$ | $K_2O$ | CaO | MgO | $Na_2O$ | A/S |
|---|---|---|---|---|---|---|---|---|---|
| BCFD/bauxite | 69.43 | 13.03 | 7.83 | 3.98 | 0.77 | 0.05 | 0.39 | 0.45 | 5.33 |
| Desiliconized | 75.35 | 6.43 | 6.90 | 3.75 | 0.42 | 0.02 | 0.33 | 0.43 | 11.72 |

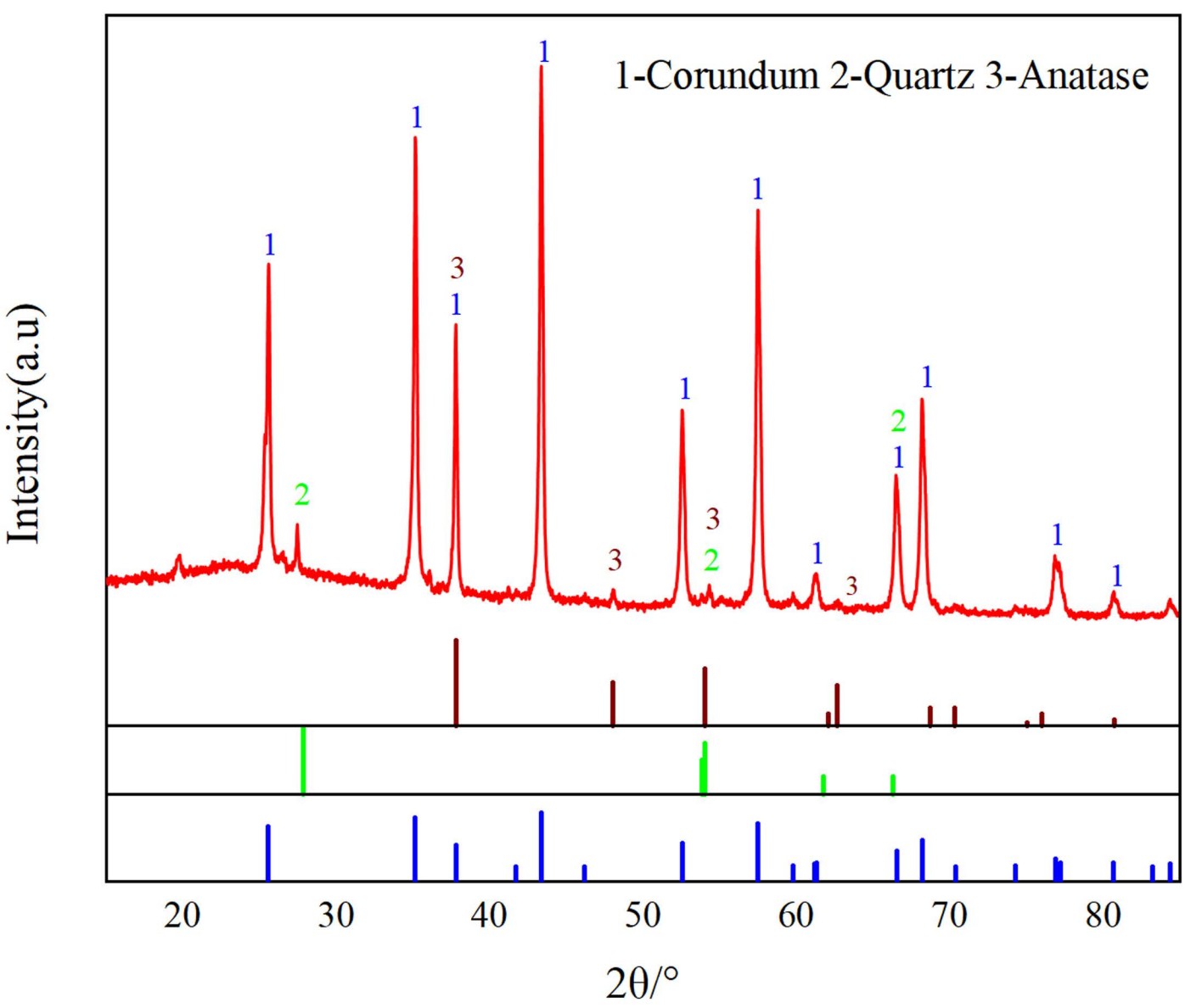

**Fig 7. Desilication concentrate XRD.**

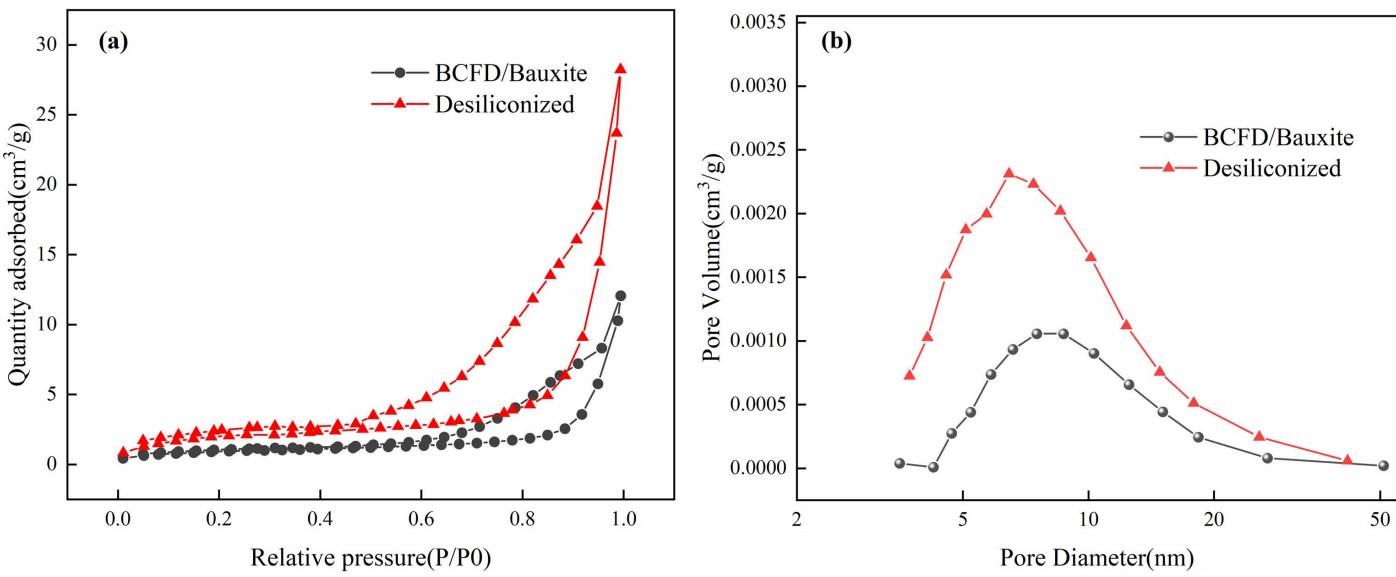

**Fig 8. Particle size distribution before and after desiliconization of mixed ore** (a), absorption-desorption curve (b), pore size distribution.

controlled by internal diffusion and chemical reaction [30]. The chemical reaction control kinetic equation is:

$$t_1 = k[1-(1-n)^{1/3}] \tag{4}$$

where $t_1$ is the leaching time (s); n is the desilication rate; k is the chemical reaction rate constant.

The internal diffusion control kinetic equation is:

$$t_2 = k[1-2n/3-(1-n)^{2/3}] \tag{5}$$

where $t_2$ is the leaching time (s); n is the desilication rate; k is the chemical reaction rate constant.

The relationship between the desilication effect and time at different temperatures was fitted with the chemical reaction control kinetic equation (4), and as shown in Fig 8(a) the correlation coefficient was found to be greater than 0.846. Subsequently, the internal diffusion control kinetic equation (5) was fitted, and as shown in Fig 8(b) the $R^2$ values were all greater than 0.97, indicating that the shrinking core model $1-2/3n-(1-n)^{2/3}$ is linearly related to time, suggesting that the reaction is controlled by internal diffusion. The unreacted core shrinkage model is related to the desilication time, from which the reaction constant k can be obtained. According to the Arrhenius equation $\ln k = \ln A - Ea/(RT)$, the plots of $1/T$ and $\ln k$ at different temperatures was drawn, as shown in Fig 8(c), indicating that the relationship between $\ln k$ and $1/T$ is linear, with the slope equal to $-Ea/R$. The desilication reaction has a better linear relationship, which is fitted with $R^2$ as a 0.994 of a straight line. Therefore, based on the slope of the line, the activation energy at a desilication temperature of 75°C to 95°C can be calculated as 29.30 kJ/mol (Fig 9).

Therefore, the synergistic desilication process of BCFD and bauxite belongs to the internal diffusion control in the unreacted core shrinkage model, and the desilication kinetic equation is:

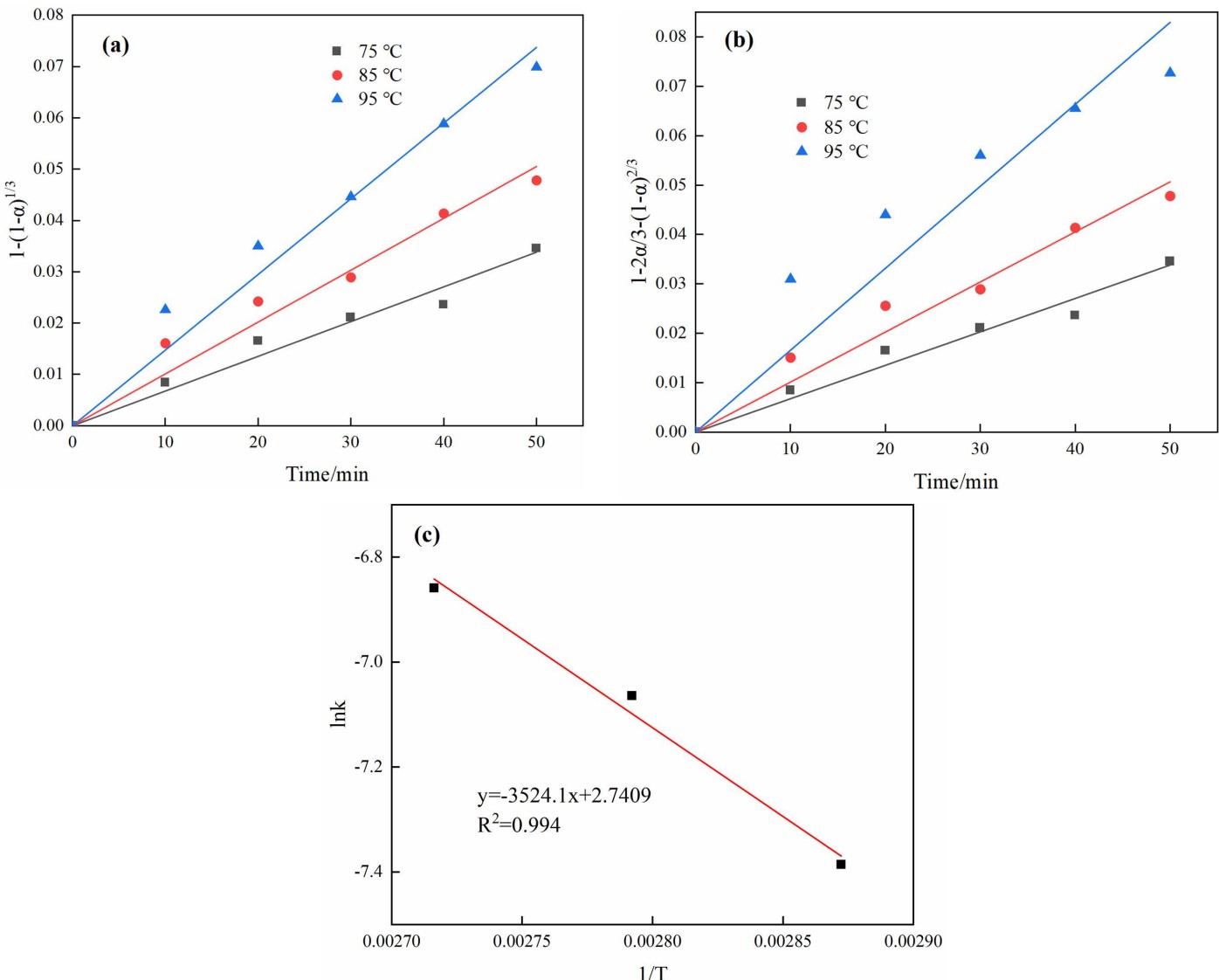

**Fig. 9. Fitting of desilication kinetics** (a) Fitting of chemical reaction kinetics, (b) Fitting of internal diffusion kinetics; (c) Arrhenius plot of the desilication react.

$$1-2/3\alpha-(1-\alpha)^{2/3} = 15.50\exp\left[-29299/(RT)\right]t.$$

## 3.3 Desilication mechanism

After roasting, the bauxite is activated, and the ore becomes loose, which is beneficial for subsequent desilication. The desilication effect of BCFD and bauxite separately is not satisfactory. The fine BCFD particles are enriched on the bauxite particles, resulting in a strong interaction and a seeding effect. After mixing with the ore, the mineral surface area increases, and during the alkaline leaching process, the reaction is influenced by internal diffusion. The amorphous silica in the mixed ore preferentially enters the alkali solution, and together with the subsequent entry of aluminum, it attaches to the surface of the seeds, significantly enhancing the desilication activity and increasing the desilication rate of the mixed minerals.

## 4. Conclusion

To improve the utilization of BCFD and reduce impurities in the raw materials for brown corundum, a "roasting-alkaline leaching" method was proposed for the mixed desilication of BCFD and bauxite, and the influence of desilication conditions and the mechanism of BCFD were investigated. The main conclusions are as follows:

1. The optimum conditions were raw material ratio of 1:6, desilication temperature and time of 95°C and 30 min, alkali concentration and liquid-solid ratio of 110 g/L and 10:1, and the desilication rate could reach 60.90%. After desilication, the mineral A/S increased from 5.33 to 11.72, and the desilicated concentrate has reached the requirement of brown corundum smelting raw material.

2. BCFD can increase the mineral surface and the amount of active silica after mixing due to its fine particle size, which can speed up the reaction rate.

3. The desilication reaction conforms to the solid shrinkage nucleation model, with the kinetic equation $1-2/3\alpha-(1-\alpha)^{2/3}=15.50\exp[-29299/(RT)]t$, apparent activation energy 29.30 kJ/mol, indicating that the process is controlled by internal diffusion. This study can apply the lower grade bauxite and BCFD to reduce the waste of resources and increase the choice of raw materials for brown corundum smelting.

## Supporting information

**S1 File. X-ray diffraction data of different minerals in** Fig 1**.** a, b represent bauxite and BCFD, respectively.
(ZIP)

**S2 File. X-ray diffraction data for different ratios of desilicated in** Fig 3(a)**.** a, b, c, d, e, and f correspond to the samples with ratios of 1:0, 1:2, 1:4, 1:6, 1:8, and 0:1, respectively.
(ZIP)

## Acknowledgments

We thank the research School of Materials and Metallurgy, Guizhou University and Guizhou Key Laboratory of Metallurgical Engineering and Process Energy Conservation for their comments on an earlier draft of this manuscript.

## Author contributions

**Data curation:** Nian Liu, Dong Liang.

**Formal analysis:** Nian Liu, Dong Liang.

**Supervision:** Chaoyi Chen, Junqi Li.

**Writing – original draft:** Nian Liu.

**Writing – review & editing:** Chaoyi Chen, Junqi Li.

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
