## [Decision Letter · Decision Letter 0]

31 Jul 2024

PONE-D-24-27758Research on desiliconization of brown corundum fly dust and bauxite based on roasting-alkali leaching methodPLOS ONE

Dear Dr. Liu,

Thank you for submitting your manuscript to PLOS ONE. After careful consideration, we feel that it has merit but does not fully meet PLOS ONE’s publication criteria as it currently stands. Therefore, we invite you to submit a revised version of the manuscript that addresses the points raised during the review process.

We look forward to receiving your revised manuscript.

Kind regards,

Mahmood Ahmed

Academic Editor

PLOS ONE

Journal Requirements:

5. Please amend the manuscript submission data (via Edit Submission) to include author Chaoyi Chen.

Reviewers' comments:

Reviewer's Responses to Questions

**Comments to the Author**

1. Is the manuscript technically sound, and do the data support the conclusions?

Reviewer #1: Yes

Reviewer #2: Yes

2. Has the statistical analysis been performed appropriately and rigorously? 

Reviewer #1: Yes

Reviewer #2: Yes

3. Have the authors made all data underlying the findings in their manuscript fully available?

Reviewer #1: Yes

Reviewer #2: Yes

4. Is the manuscript presented in an intelligible fashion and written in standard English?

Reviewer #1: Yes

Reviewer #2: Yes

5. Review Comments to the Author

Reviewer #1: Manuscript ID: PONE-D-24-27758

Title: Research on desiliconization of brown corundum fly dust and baux2 ite based on roasting-alkali leaching method

Authors: Nian Liu et al.

I cannot find keywords; Authors should improve add this information.

Line 48-64. There is no detailed description of the methods with indication of technological parameters (temperature, concentration of reagents, duration of the process and metals extraction degree). It is recommended that the authors add numerical values to the text of this section.

Table 1. Add the LOI content, what is the CaO, MgO, Na2O content? Add the carbon and sulphur content for the BCFD sample.

Authors should use “min”, “desilication”, not “minutes” and “desiliconization”.

Section 2.1. Add information about particle size distribution of the raw materials.

Table 2. Add the LOI, CaO, MgO content.

Add the information about chemical composition (g/L) after alkali leaching. Write the options for treatment this solution using the links.

Add information about particle size distribution of the sample after desilication.

What is the yield of the sample after roasting, and after desilication? Add this information to the article text.

Figure 1. Add the chemical formulas of the minerals.

Figure 3. Add the error bars to the columns and experimental points.

Figure 3d, 7. Sign peak at 20 degrees.

Figure 8. A very common mistake was made in the calculations. The curves do not pass-through point 0, it simply is not there. This is a mistake; you need to redo the calculation. You can read more information here: Levenspiel O. Chemical Reaction Engineering. https://the-seventh-dimension.com/images/textlev/LEVENSPIEL%20Chemical%20reaction%20engineering-ch1-ch2.pdf

Authors should add the final flowsheet with the optimum technical parameters.

Reviewer #2: This paper used low-grade bauxite mixed with solid waste (Brown corundum fly dust) for desilication to obtain a desilicated concentrate with a high alumina-to-silica ratio. It not only avoids the problem of solid waste stockpiling, but also makes the solid waste and low-quality bauxite turn into high-quality raw materials with high alumina-to-silica ratios, which has a good research value. However, the article still suffers from the following problems and revisions are recommended.

1. Picture clarity is low, need to improve picture quality.

2. It is suggested that some expressions be modified, for example, in line 117, “As shown in Fig 4(c)(d),” it is suggested that it be modified to read “”As shown in Fig 4(c) and (d)“”.

3. In line 69, the two sentences are not separated. For example:“Ding [21] treated the BCFD with ultrasound, and when the sulfuric acid concentration was 25 wt%, the gallium leaching rate was 82.56% Xiong [22] proposed .......”，It should be modified to：““Ding [21] treated the BCFD with ultrasound, and when the sulfuric acid concentration was 25 wt%, the gallium leaching rate was 82.56% . Xiong [22] proposed ......”

4. There is a sign error in line 204, for example:“solubility of alumina.[26, 27]. ”Suggested revision to“solubility of alumina [26, 27].”

5. Why choose 1050℃ for roasting activation temperature?

6. The desilication product XRD pattern suggests the addition of a standard card.

6. PLOS authors have the option to publish the peer review history of their article (what does this mean? ). If published, this will include your full peer review and any attached files.

**Do you want your identity to be public for this peer review?** For information about this choice, including consent withdrawal, please see our Privacy Policy .

Reviewer #1: No

Reviewer #2: **Yes: ** Junhui Xiao

---

## [Author Response · Author response to Decision Letter 0]

26 Sep 2024

Thank you for providing the thoughtful and constructive comments from the reviewers concerning our manuscript titled “Research on Desiliconization of Brown Corundum Fly Dust and Bauxite Based on Roasting-Alkali Leaching Method.” We have carefully considered each of the comments and have made comprehensive revisions to our manuscript. Below, we detail the changes made in response to each point raised by the reviewers.

We hope that these revisions satisfactorily address the concerns raised by the reviewers and enhance the manuscript’s contribution to the field. We appreciate the reviewers’ insights and thank the editorial team for their guidance.

---

## [Editor Report · Decision Letter 1]

2 Oct 2024

PONE-D-24-27758R1Research on desiliconization of brown corundum fly dust and bauxite based on roasting-alkali leaching methodPLOS ONE

Dear Dr. Liu,

Thank you for submitting your manuscript to PLOS ONE. After careful consideration, we feel that it has merit but does not fully meet PLOS ONE’s publication criteria as it currently stands. Therefore, we invite you to submit a revised version of the manuscript that addresses the points raised during the review process.

**ACADEMIC EDITOR: **

**One or more of the reviewers has recommended that you cite specific previously published works. Members of the editorial team have determined that the works referenced are not directly related to the submitted manuscript. As such, please note that it is not necessary or expected to cite the works requested by the reviewer**

We look forward to receiving your revised manuscript.

Kind regards,

Mahmood Ahmed

Academic Editor

PLOS ONE

---

## [Author Response · Author response to Decision Letter 1]

10 Oct 2024

Thanks to the comments of the academic editors, I have carefully revised the suggestions that have been made. I apologize for the inconvenience caused by the addition and deletion of some citations in the manuscript, as there are few relevant studies in this area, the relevance of the listed documents meets the requirements as much as possible, and the suggestions made by the editors have been carefully verified and dealt with.

We believe these revisions have addressed the concerns raised by the reviewers and have significantly improved the manuscript. We appreciate your consideration of our revised submission and eagerly await further feedback. Thank you for considering our responses, and we look forward to the potential acceptance and publication of our manuscripts.

---

## [Decision Letter · Decision Letter 2]

8 Dec 2024

Research on desiliconization of brown corundum fly dust and bauxite based on roasting-alkali leaching method

PONE-D-24-27758R2

Dear Dr. Chen,

We’re pleased to inform you that your manuscript has been judged scientifically suitable for publication and will be formally accepted for publication once it meets all outstanding technical requirements.

Kind regards,

Mahmood Ahmed

Academic Editor

PLOS ONE

Additional Editor Comments (optional):

Accept

Reviewers' comments:

Reviewer's Responses to Questions

**Comments to the Author**

1. If the authors have adequately addressed your comments raised in a previous round of review and you feel that this manuscript is now acceptable for publication, you may indicate that here to bypass the “Comments to the Author” section, enter your conflict of interest statement in the “Confidential to Editor” section, and submit your "Accept" recommendation.

Reviewer #2: All comments have been addressed

Reviewer #3: All comments have been addressed

Reviewer #4: All comments have been addressed

Reviewer #5: All comments have been addressed

2. Is the manuscript technically sound, and do the data support the conclusions?

Reviewer #2: Yes

Reviewer #3: Yes

Reviewer #4: Yes

Reviewer #5: Yes

3. Has the statistical analysis been performed appropriately and rigorously? 

Reviewer #2: Yes

Reviewer #3: Yes

Reviewer #4: Yes

Reviewer #5: Yes

4. Have the authors made all data underlying the findings in their manuscript fully available?

Reviewer #2: Yes

Reviewer #3: Yes

Reviewer #4: Yes

Reviewer #5: Yes

5. Is the manuscript presented in an intelligible fashion and written in standard English?

Reviewer #2: Yes

Reviewer #3: Yes

Reviewer #4: Yes

Reviewer #5: Yes

6. Review Comments to the Author

Reviewer #2: The authors have answered my questions and the quality of the manuscript impoved a lot. My recommendation is accept.

Reviewer #3: The author revised the manuscript very well according to the comments, the paper can be published in present submission.

Reviewer #4: (No Response)

Reviewer #5: (No Response)

7. PLOS authors have the option to publish the peer review history of their article (what does this mean? ). If published, this will include your full peer review and any attached files.

**Do you want your identity to be public for this peer review?** For information about this choice, including consent withdrawal, please see our Privacy Policy .

Reviewer #2: **Yes: ** Junhui Xiao

Reviewer #3: No

Reviewer #4: No

Reviewer #5: No

---

## [Editor Report · Acceptance letter]

PONE-D-24-27758R2

PLOS ONE

Dear Dr. Chen,

I'm pleased to inform you that your manuscript has been deemed suitable for publication in PLOS ONE. Congratulations! Your manuscript is now being handed over to our production team.

Kind regards,

on behalf of

Dr. Mahmood Ahmed

Academic Editor

PLOS ONE